# Measuring the Wellbeing of Cancer Patients with Generic and Disease-Specific Instruments

**DOI:** 10.3390/cancers15041351

**Published:** 2023-02-20

**Authors:** Gang Chen, Norma B. Bulamu, Ellen McGrane, Jeff Richardson

**Affiliations:** 1Centre for Health Economics, Monash University, Melbourne, VIC 3145, Australia; 2Flinders Health and Medical Research Institute, Flinders University, Adelaide, SA 5042, Australia; 3School of Health and Related Research, University of Sheffield, Sheffield S1 4DA, UK

**Keywords:** cancer, health state utility, subjective wellbeing, capability, life satisfaction

## Abstract

**Simple Summary:**

Patient-reported outcomes play an important role in clinical trials and health economic evaluation. In addition to health-related quality of life, there has been increasing recognition of measuring wider wellbeing that goes beyond health. This study aims to understand the sensitivity and comparability of commonly used preference-based health-related quality of life and subjective wellbeing measures in patients with cancer. This study further explored the life domain importance of cancer patients. The findings from this study shed light on the choice of patient-reported outcome measures in clinical studies as well as the prioritized aspects to improve the overall life satisfaction of cancer patients.

**Abstract:**

Different wellbeing measures have been used among cancer patients. This study aimed to first investigate the sensitivity of health state utility (HSU), capability, and subjective wellbeing (SWB) instruments in cancer. A cancer-specific instrument (QLQ-C30) was included and transferred onto the cancer-specific HSU scores. Furthermore, it examined the relative importance of key life domains explaining overall life satisfaction. Data were drawn from the Multi-instrument Comparison survey. Linear regression was used to explore the extent to which the QLQ-C30 sub-scales explain HSU and SWB. Kernel-based Regularized Least Squares (KRLS), a machine learning method, was used to explore the life domain importance of cancer patients. As expected, the QLQ-C30 sub-scales explained the vast majority of the variance in its derived cancer-specific HSU (R^2^ = 0.96), followed by generic HSU instruments (R^2^ of 0.65–0.73) and SWB and capability instruments (R^2^ of 0.33–0.48). The cancer-specific measure was more closely correlated with generic HSU than SWB measures, owing to the construction of these instruments. In addition to health, life achievements, relationships, the standard of living, and future security all play an important role in explaining the overall life satisfaction of cancer patients.

## 1. Introduction

Generic health state utility (HSU) measures have been widely used to measure one concept of health-related quality of life (HRQoL). They allow decision-makers to generate quality-adjusted life years (QALYs) and are employed in cost-utility analysis for prioritizing health services. QALY is a metric that combines the quantity of life experienced in a particular health state and the value individuals place on that life [1]. Generic HSU measures consist of a descriptive system (a set of questions describing the respondent’s health state) and a scoring algorithm that converts the answers into a utility score anchored on a zero (death) to one (best health) QALY scale [1].

Within the literature, some commentators have argued that generic HSU measures may not be sensitive to the mix of symptoms that are unique to a particular disease, as measured by a disease-specific instrument [2,3,4]. To overcome this problem, disease-specific HSU instruments have been created, in which some or all of the responses to a disease-specific instrument are weighted to obtain an overall disease-specific HSU. The resulting instruments may appear to be a gold standard for cost-utility analyses, as they combine a sensitive descriptive system with a measure of the relative importance of its components.

While cost-utility analyses assume that effectiveness should be measured by utility, another study focused on the measurement of subjective wellbeing (SWB), which is defined (depending on the instrument) by a person’s satisfaction, happiness, or eudaimonia (a sense of purpose/meaning in life) [5]. Similar to generic HSU measures, SWB measures have also been used across different diseases, including cancer [6]. On the other hand, SWB measures differ from generic measures of health, which may fail to pick up important non-health aspects of life for cancer patients; consequently, the use of SWB instruments alongside HRQoL may further improve our understanding of the longer-term effects of cancer on overall wellbeing [6]. Indeed, cancer patients are at an increased risk of social isolation as a consequence of poor health, reduced ability to function at work or early retirement [7], and changes in peer and family relationships [8]. People living with cancer experience various forms of social difficulty [9] that negatively affect patients’ social engagement, social identity, and social networks and may ultimately lead to a reduction in the quantity and quality of social support they receive [10]. Recent literature has also developed instruments for measuring a patient’s capabilities, a concept promoted and developed by Amartya Sen [11]. The development of capability wellbeing facilitates the economic evaluation of a broader wellbeing outcome rather than health [12].

In this paper, we aim to explore and compare the relationships between cancer-specific instruments and wellbeing as measured by HSU, SWB, and capabilities. More specifically, we first aim to understand which generic HSU instruments are more sensitive to quality-of-life domains as measured by a widely used cancer-specific instrument. The results shed light on the choice of benefit measures for cancer patients. Secondly, we explore the relative importance of key life domains in explaining global life satisfaction among cancer patients that goes beyond health.

## 2. Materials and Methods

### 2.1. Respondents

Data were obtained from a Multi-instrument Comparison Study of 8022 individuals in 6 countries [13,14]. The survey was administered to patients diagnosed with 1 of 7 chronic illnesses. Respondents (N = 772) who reported having cancer were used in this study.

### 2.2. Instruments

The study involved three SWB, one capability-wellbeing, and six generic HSU instruments and one cancer-specific instrument. A disease-specific HSU instrument was further derived from the cancer-specific instrument. The dimensions of HRQoL included in the instruments are compared in Table 1.

#### 2.2.1. Cancer-Specific Instruments

The EORTC QLQ-C30 contains five multi-item functional scales (role, physical, cognitive, emotional, and social functioning), three multi-item symptom scales (fatigue, pain, and nausea and vomiting), five single-item symptoms scales (diarrhea, appetite loss, dyspnea, constipation, and insomnia), one item on financial difficulties, and two questions assessing global health status / overall quality of life [15]. It is scored on a four-point scale (1 = ‘Not at all’, 2 = ‘A little’, 3 = ‘Quite a bit’, 4 = ‘Very much’) [23]. It is validated and widely used in the cancer literature [24]. The EORTC Quality of Life Utility Measure-Core 10 Dimensions (QLU-C10D) [22] is a preference-based HSU instrument derived from the QLQ-C30. Utility scores for QLU-C10D are obtained from responses to 10 dimensions of the QLQ-C30 (physical, role, social, and emotional functioning; pain, fatigue, sleep, appetite, nausea, and bowel problems) [22]. A value set for the QLU-C10D obtained from an Australian general population sample was applied in this study [25].

#### 2.2.2. Subjective Wellbeing and Capability Wellbeing Instruments 

The three SWB instruments included the Office for National Statistics 4 (ONS4), the Personal Wellbeing Index (PWI), and the Satisfaction with Life Scale (SWLS), while the capability wellbeing was measured using the ICEpop CAPability measure for Adults (ICECAP-A). The ONS4 was introduced into the UK Integrated Household Survey by the ONS in 2011 [26]. It is a set of 4 questions that measure SWB across four domains—satisfaction with life, eudaimonia, and positive and negative effect—each with 11 response categories from ‘not at all’ to ‘completely’. The PWI assesses respondents’ satisfaction with 7 life domains and an optional satisfaction question on life as a whole (i.e., ‘Thinking about your own life and personal circumstances, how satisfied are you with your life as a whole?’) using 11 response categories from 0 to 10 [27,28]. These include the standard of living, personal health, life achievements, personal relationships, personal safety, feeling part of the community, and future security. The SWLS assesses global life satisfaction with 5 items that focus on present and past life (e.g., ‘In most ways my life is close to my ideal’, and ‘If I could live my life over, I would change almost nothing’). It is scored on a 7-point Likert scale from ‘strongly disagree’ to ‘strongly agree’ and has a total score ranging from 5 to 35 [29]. Both SWLS and PWI have been validated to measure the SWB in cancer patients [30,31,32]. All three SWB scores were rescaled onto a 0–1 scale in the empirical analysis. The ICECAP-A differs from these, as it is preference-based and was developed based on the capability approach [33]. It is measured across five attributes/domains; stability (‘feeling settled and secure’), attachment (‘love, friendship, and support’), autonomy (‘being independent’), achievement (‘achievement and progress’), and enjoyment (‘enjoyment and pleasure’). Each attribute has four levels, and the instrument is anchored on a 1 (full capability) to 0 (no capability) scale.

#### 2.2.3. Generic Health State Utility Instruments

The six generic HSU instruments used in this study were the 15D, AQoL-8D, EQ-5D-5L, HUI3, SF-6D, and SF-6Dv2. Except for the newly developed SF-6Dv2, all instruments are described and contrasted in Richardson et al. [13]. The 15D measures 15 dimensions (each with 5 levels) of HRQoL: mobility, vision, hearing, breathing, sleeping, eating, speech, elimination, usual activities, mental function, discomfort and symptoms, depression, distress, vitality, and sexual activity [16]. The AQoL-8D has the broadest descriptive system, with 35 items grouped into 8 dimensions, namely independent living, happiness, mental health, coping, relationships, self-worth, pain, and senses [17]. The EQ-5D is one of the most widely used and validated HSU instruments that measure HRQoL according to five dimensions: mobility, self-care, usual activities, pain/discomfort, and anxiety/depression [34]. The five-level EQ-5D (EQ-5D-5L) is a newer version of the instrument that comprises five levels for each of the dimensions (no, slight, moderate, severe, and extreme problems) [18,35]. The HUI3 was developed in Canada and has 8 dimensions: vision, hearing, speech, ambulation, dexterity, emotion, cognition, and pain, each having 5–6 levels [19]. The SF-6D is a preference-based HSU instrument that was developed based on 11 items of the Short Form-36, which was designed to measure 6 dimensions of HRQoL: physical functioning, role limitation, social functioning, pain, mental health, and vitality [20]. SF-6Dv2 is a new instrument that was developed based on 10 items of SF-36 to overcome some limitations of SF-6D [21,36]. By far, limited empirical evidence is available regarding its relative performance against the original SF-6D [37].

### 2.3. Statistical Analyses

Descriptive statistics, including Spearman correlation coefficients, have been calculated. The percentages of respondents to be classified as in the best (full) and worst health/wellbeing state using each instrument were reported. The sensitivity of different generic measures to the dimensions and symptoms in the QLQ-C30 was tested using an ordinary least-squares (OLS) model with a forward stepwise selection technique to choose the statistically significant regressors (with a significance level of 5% to be included in the final model). The QLU-C10D, derived from the QLQ-C30, was also included to serve as a comparison. The use of stepwise regression facilitated a clearer comparison of significant regressors across different wellbeing outcomes. In Appendix A Table A1, the regression analyses, including all QLQ-C30 scales as well as other covariates (a set of age, education, and country dummies), are presented. The standardized beta coefficients are reported to facilitate comparisons of relative importance.

Life domain importance was studied following the bottom-up framework that global life satisfaction is explained by life domain satisfaction [38]. SWLS was used as the global life satisfaction measure, while the seven key life domain satisfaction comes from PWI. To account for the potential non-linear effect from life domains, this study adopted a more flexible machine learning method, Kernel-based Regularized Least Squares (KRLS), which relaxes the linearity or additivity assumptions [39,40]. Similarly, to enable the comparisons of relative importance between life domains, all life satisfaction variables were first standardized before entering the regression analyses. All the statistical analyses were performed using Stata 15.0 (StataCorp LP, College Station, TX, USA).

## 3. Results

### 3.1. Respondent Characteristics

The study sample comprised 772 self-reported cancer survivors (54% female, 67% aged ≥55 years old). The mean SWB scores of ONS4, PWI, and SWLS (varying from 0.56 to 0.63) were consistently lower than the mean capability wellbeing (ICECAP-A) score (0.81). The mean HSU scores ranged from 0.63 (SF-6Dv2) to 0.82 (15D). Based on the descriptive system of each instrument, 12% of the respondents were classified to be in full health according to EQ-5D-5L, while 10% of the respondents were classified to be in full capability (ICECAP). For all other instruments, less than 5% of the respondents were classified to be in full health/wellbeing. On the other hand, less than 2% of the respondents were classified to be in the worst health/wellbeing for all instruments. For more descriptive statistics, see Table 2.

### 3.2. Correlations between Cancer-Specific and Wellbeing Instruments

Table 3 shows the significant correlations between subscales of the QLQ-C30 and the SWB, capability, and HSU measures. As expected, all correlations with the QLQ-C30 functional subscales were positive, while correlations with the symptom subscales were negative, and all measures were highly correlated with the global score. Strengths of correlations between QLQ-C30 functional subscales and the wellbeing measures ranged from 0.26 (role functioning and SWLS) to 0.60 (emotional functioning and ONS4). The highest and lowest correlations for all wellbeing measures were with emotional-functioning and role-functioning subscales, respectively, excluding the ICECAP-A, where the lowest correlation was with cognitive functioning. Correlations with the HSU measures ranged from 0.41 (cognitive functioning and EQ-5D-5L) to 0.83 (physical functioning and QLU-C10D). The highest and lowest correlations for all HSU measures were with physical and cognitive functioning, respectively, excluding the AQoL-8D, where the highest correlation (0.69) was with emotional functioning.

Among QLQ-C30 symptom scales, the lowest correlation between wellbeing measures was 0.14 (diarrhea and SWLS), and the highest was 0.53 (fatigue and ICECAP-A). The lowest correlation for all wellbeing measures was with diarrhea, excluding the ONS4 (0.21), which was with constipation. Correlations with the HSU measures ranged from 0.25 (diarrhea and HUI3) to 0.81 (fatigue and QLU-C10D). The lowest correlation for all utility measures was with diarrhea, but the highest correlations varied between pain (EQ-5D-5L, SF-6Dv2, and HUI3) and fatigue (QLU-C10D, 15D, AQoL-8D, SF-6D, and SF-6Dv2).

All measures were strongly correlated with the QLQ-C30 global quality of life score (range: 0.60–0.75). It is not surprising that the strongest correlation between QLQ-C30 and HSU measures was QLU-C10D (except for emotional, cognitive functioning, and dyspnea), as these measures are derived from the QLQ-C30.

### 3.3. Explanation of Generic Instruments Using QLQ-C30 Subscales

Regression analyses shown in Table 4 found that, among three groups of instruments, the exploratory power of the functional and symptom subscales was greatest for the HSU instruments (R^2^ between 0.66 to 0.96), followed by the capability instrument (R^2^ = 0.50) and the SWB instruments (R^2^ between 0.34 to 0.46). Among QLQ-C30 subscales, emotional functioning had a significant impact on all wellbeing and HSU measures (the largest coefficient was with the ONS4). Other subscales that significantly impacted the wellbeing measures were financial difficulties (largest coefficient with SWLS), insomnia (largest coefficient with ONS4), physical functioning (largest coefficient with ICECAP-A), Pain (largest coefficient with SWLS), and dyspnea (significant with PWI).

The QLU-C10D was sensitive to the largest number of subscales (11 out of 14), which was as expected given it is derived directly from the QLQ-C30. The sensitivity of generic HSU to subscales varied with the instrument used. The largest coefficients for emotional functioning and financial difficulties scales were obtained with the AQoL-8D; that for the physical functioning was with the HUI3; and those for the cognitive functioning, dyspnea, insomnia, and constipation were with the 15D.

### 3.4. Relative Importance of Key Life Domains

Table 5 first shows the descriptive statistics (mean and SD) of seven life domain satisfaction variables. Among them, respondents had the highest mean score on personal safety, followed by personal relationships, the standard of living, community connectedness, achieving in life, and future security, while scoring the lowest on health. Next, the KRLS estimates on life domain importance are presented. In Column (1), except for personal safety, all life domains were statistically significant. Based on the magnitudes of the estimations, life achievements was the most important life domain for overall life satisfaction, followed by personal relationships, the standard of living, future security, personal health, and community connectedness. Column (2) further includes the QLU-C10D in the regression. The inclusion of QLU-C10D (which was insignificant) had a minimum influence on the results in Column (1), except for reducing the estimate of the personal health domain slightly. Column (3) includes ICECAP-A to replace QLU-C10D, and it was found to be significant. The inclusion of ICECAP-A reduced the estimates of all seven life domains and improved the variances of overall life satisfaction explained.

## 4. Discussion

Using cross-sectional data from patients who self-reported cancer in six developed countries, this study investigated the extent to which patients’ SWB and generic and cancer-specific HSUs are sensitive to the functional and symptom subscales employed by the QLQ-C30 cancer-specific instrument. Among the broad categories of instruments, it was found that the QLQ-C30 subscales are more closely related to utility scores than SWB scores. The correlation with cancer-specific scores was higher than that with the scores from generic instruments. These results are reflected in the linear regressions, where the R^2^ statistics (in Table 4) are lowest when the dependent variable was SWB (0.33 to 0.48) and higher for the generic HSU (0.65 to 0.73). While the R^2^ statistics on the cancer-specific HSU is the largest (0.96), this is primarily due to their etiology. The results are unsurprising, as the utility instruments measure similar concepts to the QLQ-C30 (see Table 1) and the cancer-specific HSUs are derived from the QLQ-C30. The relative importance of subscales also differs. From Table 4, SWB was most affected by emotional functioning, while the utility was most affected by physical functioning.

The much weaker correlation between cancer-specific measures and SWB measures as compared to HSU measures may be attributed to their broader content. All SWB measures are sensitive to the financial difficulties subscale, while both the most widely generic HSUs (EQ-5D-5L and SF-6D) are not. This becomes clear in Table 5, in which very limited influence is found when adding QLU-C10D into the life domain importance model; i.e., cancer-specific HSU does not cover the potential influence of broader life domains on overall wellbeing in addition to health. When capability wellbeing is included in the life domain importance model, as expected, more variance of overall life satisfaction is explained. It can be argued that SWB may not be interchangeable with HRQoL when considering outcomes in cancer [6]. These results indicate that the importance of the different subscales, both between and within the three categories of instruments, depends, to a large extent, on the construction of these instruments, a result found more generally by Khan and Richardson [41].

A less expected result is the variable strength of the association of QLQ-C30 subscales by instruments within conceptually similar subgroups. From Table 4, SWB is not sensitive to pain when measured by the PWI but is sensitive to dyspnea. The converse is true for the ONS4 and SWLS, which are sensitive to pain but not to dyspnea. Between SF-6D and SF-6Dv2 (both of which were derived from SF-36), the original SF-6D is sensitive to seven subscales, while the recently developed SF-6Dv2 is sensitive to five subscales, with the differences mainly occurring in social functioning, fatigue, insomnia, and financial difficulties. Similar differences exist between generic HSU instruments. While emotional functioning had the largest effect on the AQoL-8D and SF-6D, other instruments are more sensitive to physical functioning (15D, HUI3) or pain (EQ-5D-5L, SF-6Dv2). Four symptom scales affect the 15D and AQoL-8D; three affect the HUI3 and SF-6D, and two affect SF-6Dv2 and EQ-5D-5L. The different performance reported here implies that the choice of instruments can lead to different economic evaluation conclusions of cancer-related trials or interventions (which is particularly the case given the important role played by the generic HSU instruments in cost-utility analyses).

The life domain analyses revealed that the top three most important domains are life achievements, personal relationships, and the standard of living (with standardized coefficients all close to or above 0.18 as compared to the fourth life domain of future security with a coefficient of 0.12). These three life domains have been commonly found to be among the top four important life domains for overall life satisfaction [42,43,44]. The finding of a strong influence of the relationships domain in cancer patients is consistent with the literature, which found that social participation and strong personal relationships help individuals suffering from illness recover more quickly and adjust better [45,46,47,48,49]. As compared to patients with other chronic diseases (e.g., heart diseases) or the general public, the health domain ranks relatively lower in cancer patients in this study [43,44]. This finding differs from the study that asked cancer patients to directly rate the importance of key life domains (in which health was the highest ranked) [50]. Potential reasons for the differences include that, although cancer patients can rate health as the most important domain in stated preference tasks, in daily lives, they may be more likely to feel a loss of control over the treatment plan, cancer progress, and the fact that cancer is less likely to be cured than other diseases. Consequently, health (which is less in control of cancer patients themselves) is moderately ranked in the revealed preference reported in this study.

This study has a number of limitations. Firstly, cancer patients were recruited via an online panel, and there was no information regarding the type, stage, or duration of cancer. Secondly, since only cross-sectional data were used, responsiveness, test–retest reliability, and causality cannot be explored from the current analyses. With the advancements in big data research in cancer, such as the linkage of various nationwide datasets (containing both clinical and patient-reported outcomes) and the development of machine learning algorithms, researchers and clinicians will have the potential to provide timely and personalized feedback to support and improve the wellbeing of patients.

An important consideration is also the potential impact of using an alternative cancer-specific measure in this study. The Functional Assessment of Cancer Therapy-General Scale (FACT-G) is another popular measure, which has been suggested to have lower variability, greater discriminative ability [51], and higher efficiency for detecting changes in overall HRQoL [51,52]. Inconsistencies with the QLQ-C30 are particularly apparent in the social domain [52,53,54,55], where the FACT-G focuses on social support and relationships, an important aspect of the wellbeing of cancer patients, as mentioned above [52]. Therefore, the FACT-G may capture important aspects of the SWB of cancer patients that the QLQ-C30 fails to pick up, potentially strengthening the correlation with SWB measures.

## 5. Conclusions

This paper demonstrates that conceptually similar wellbeing instruments are sensitive to different cancer problems because of their construction. The QLU-C10D was the most sensitive instrument to most patient problems; the 15D was the most successful generic HSU instrument. This study also shows that to improve the overall life satisfaction of cancer patients, life domains, such as achievement in life, relationships, the standard of living, and future security, all play an important role in addition to health. The empirical evidence from this study indicates that SWB may not be interchangeable with HRQoL when considering outcomes in cancer. Considering the QLQ-C30 is widely used in clinical trials and the capability to produce the cancer-specific HSU (QLU-C10D) based on the responses, the additional benefit of including another generic HSU is limited, except for allowing comparisons in health economic evaluation. To enable a broader measure of benefits for cancer patients, the inclusion of an instrument for SWB, such as PWI, would provide more specific evaluations of key life domains that go beyond health.

## Figures and Tables

**Table 1 cancers-15-01351-t001:** Dimensions covered by health state utility instruments in relation to QLQ-C30.

QLQ-C30 Subscales [15]	15D [16]	AQoL-8D [17] ^†^	EQ-5D-5L [18]	HUI3 [19]	SF-6D/SF-6Dv2 [20,21]	QLU-C10D [22] ^‡^
Physical functioning	Mobility	Independent living	Mobility	Ambulation	Physical functioning	Physical functioning
Role functioning	Usual activities	Independent living	Self-care, usual activities	Dexterity	Role limitation	Role functioning
Emotional functioning	Depression, distress	Mental health	Anxiety/depression	Emotion	Mental health	Emotional functioning
Cognitive functioning	Mental function			Cognition		
Social functioning		Relationships			Social functioning	Social functioning
Fatigue	Vitality	Coping			Vitality	Fatigue
Nausea and vomiting						Nausea
Pain	Discomfort and symptoms	Pain	Pain/discomfort	Pain	Pain	Pain
Dyspnea	Breathing					
Insomnia	Sleeping	Mental health				Sleep
Appetite loss	Eating					Appetite
Constipation						Bowel problems
Diarrhea						Bowel problems
Financial difficulties						
*Not represented*	Vision, hearing, speech, excretion, sexual activity	Happiness, self-worth, senses		Vision, hearing, speech		

Notes: ^†^ For AQoL-8D, the 8 dimensions are listed in this table instead of 35 items. ^‡^ The QLU-C10D is derived directly from the QLQ-C30 cancer-specific instrument.

**Table 2 cancers-15-01351-t002:** Descriptive statistics, N = 772.

**Panel A: Socio-demographic characteristics**	**(N, %)**
Age	
18–44	116 (15.03)
45–54	142 (18.39)
55–64	265 (34.33)
≥65	249 (32.25)
Male	355 (45.98)
Education	
High school	228 (29.53)
Diploma or certificate or similar	283 (36.66)
University and over	261 (33.81)
Country	
Australia	154 (19.95)
Canada	138 (17.88)
Germany	115 (14.90)
Norway	80 (10.36)
UK	137 (17.75)
USA	148 (19.17)
**Panel B: Subjective wellbeing and capability wellbeing**	**(Mean, SD)**
ONS4	0.634 (0.212)
PWI	0.632 (0.198)
SWLS	0.555 (0.242)
ICECAP-A *	0.806 (0.179)
**Panel C: Health state utility**	**(Mean, SD)**
15D	0.819 (0.136)
AQoL-8D	0.662 (0.219)
EQ-5D-5L	0.704 (0.225)
HUI3	0.680 (0.276)
SF-6D	0.686 (0.133)
SF-6Dv2	0.632 (0.306)
QLU-C10D	0.653 (0.256)

* ICECAP was not administrated in Norway (*N* = 692 cancer patients); there is one missing observation in the other three subjective wellbeing measures among cancer patients (*N* = 771). Three subjective wellbeing (ONS4, PWI, and SWLS) scores were rescaled on a 0–1 scale.

**Table 3 cancers-15-01351-t003:** Spearman’s correlations between QLQ-C30 scores and wellbeing measures.

	QLQ-C30		Subjective Wellbeing				Health State Utility						
	M	SD	ONS4	PWI	SWLS	ICECAP	15D	AQoL-8D	EQ-5D-5L	HUI3	SF-6D	SF-6Dv2	QLU-C10D
Global quality of life	57.31	24.21	0.602	0.605	0.557	0.606	0.723	0.723	0.647	0.656	0.715	0.690	0.750
Functional scales													
Physical functioning	76.50	23.35	0.333	0.369	0.321	0.462	0.718	0.610	0.694	0.658	0.688	0.676	0.831
Role functioning	71.18	30.59	0.294	0.326	0.263	0.433	0.641	0.551	0.632	0.580	0.673	0.649	0.788
Emotional functioning	69.65	26.39	0.597	0.543	0.499	0.586	0.575	0.689	0.516	0.534	0.612	0.580	0.603
Cognitive functioning	77.63	25.16	0.348	0.365	0.330	0.409	0.572	0.506	0.408	0.501	0.479	0.472	0.540
Social functioning	66.26	31.09	0.395	0.417	0.368	0.509	0.643	0.608	0.581	0.570	0.663	0.624	0.741
Symptom scales													
Fatigue	39.46	27.46	−0.417	−0.440	−0.377	−0.528	−0.725	−0.676	−0.637	−0.611	−0.723	−0.713	−0.812
Nausea and vomiting	11.40	21.50	−0.307	−0.272	−0.235	−0.321	−0.403	−0.390	−0.356	−0.339	−0.411	−0.399	−0.541
Pain	34.48	31.47	−0.313	−0.331	−0.274	−0.429	−0.639	−0.621	−0.750	−0.678	−0.686	−0.733	−0.808
Dyspnea	23.92	29.27	−0.267	−0.313	−0.265	−0.359	−0.565	−0.459	−0.443	−0.405	−0.443	−0.431	−0.541
Insomnia	37.44	32.99	−0.449	−0.409	−0.377	−0.475	−0.573	−0.579	−0.483	−0.502	−0.532	−0.508	−0.580
Appetite loss	17.14	28.33	−0.373	−0.333	−0.303	−0.387	−0.479	−0.479	−0.434	−0.425	−0.506	−0.486	−0.590
Constipation	15.16	25.62	−0.209	−0.205	−0.170	−0.222	−0.345	−0.307	−0.300	−0.333	−0.329	−0.320	−0.415
Diarrhea	14.08	25.13	−0.226	−0.179	−0.143	−0.215	−0.329	−0.280	−0.270	−0.250	−0.317	−0.279	−0.408
Financial difficulties	32.17	34.84	−0.404	−0.441	−0.420	−0.472	−0.515	−0.520	−0.454	−0.468	−0.520	−0.499	−0.528

All Spearman’s correlation coefficients reported in the table are statistically significant (*p* < 0.01). M, mean; SD, standard deviation.

**Table 4 cancers-15-01351-t004:** Sensitivity of wellbeing measures in cancer-specific quality of life.

	Subjective Wellbeing				Health State Utility						
	ONS4	PWI	SWLS	ICECAP	15D	AQoL-8D	EQ-5D-5L	HUI3	SF-6D	SF-6Dv2	QLU-C10D
QLQ-C30: Functional scales											
Physical functioning	0.103 **	0.085 *	0.105 *	0.212 **	0.292 **	0.179 *	0.303 **	0.334 **	0.202 **	0.209 **	0.313 **
Role functioning									0.103 **	0.120 *	0.154 **
Emotional functioning	0.505 **	0.384 **	0.342 **	0.409 **	0.112 *	0.373 **	0.201 **	0.142 *	0.233 **	0.247 **	0.122 **
Cognitive functioning					0.184 **			0.146 **			
Social functioning					0.066 *				0.089 *		0.114 **
QLQ-C30: Symptom scales											
Fatigue									−0.153 **		−0.049 *
Nausea and vomiting											−0.086 **
Pain	0.088 **		0.100 *		−0.115 *	−0.184 **	−0.426 **	−0.284 *	−0.187 **	−0.407 **	−0.231 **
Dyspnea		−0.065 *			−0.155 **	−0.094 *					
Insomnia	−0.140 **	−0.086 **	−0.112 **	−0.108 *	−0.166 **	−0.162 **	−0.047 *	−0.083 *	−0.072 *		−0.062 **
Appetite loss											−0.045 *
Constipation											−0.071 **
Diarrhea											−0.060 **
Financial difficulties	−0.111 **	−0.181 *	−0.234 **	−0.163 **	−0.063 *	−0.107 **		−0.050 **		−0.038 **	
N	771	771	771	692	772	772	772	772	772	772	772
R^2^	0.435	0.391	0.326	0.483	0.728	0.684	0.652	0.657	0.680	0.686	0.957

Statistically significant (via a stepwise selection process) and standardized beta coefficients are reported. ** *p* < 0.01, * *p* < 0.05.

**Table 5 cancers-15-01351-t005:** Life domain importance among patients with cancer.

			Model (1)		Model (2)		Model (3)	
	M	SD	Average ^†^	SE	Average ^†^	SE	Average ^†^	SE
Life Domains								
Standard of living	0.649	0.245	0.179	(0.020) **	0.180	(0.019) **	0.170	(0.020) **
Personal health	0.489	0.269	0.098	(0.017) **	0.088	(0.018) **	0.063	(0.017) **
Achieving in life	0.623	0.253	0.247	(0.020) **	0.241	(0.020) **	0.224	(0.020) **
Personal relationships	0.707	0.263	0.184	(0.019) **	0.185	(0.018) **	0.163	(0.018) **
Personal safety	0.721	0.235	0.006	(0.020)	0.004	(0.020)	−0.023	(0.020)
Community connectedness	0.645	0.243	0.042	(0.019) *	0.044	(0.019) *	0.039	(0.019) *
Future security	0.591	0.275	0.122	(0.020) **	0.117	(0.019) **	0.091	(0.020) **
Health utility (QLU-C10D)					0.027	(0.017)		
Capability wellbeing (ICECAP-A)							0.180	(0.020) **
Covariates			√		√		√	
N	771		771		771		691	
R^2^			0.780		0.782		0.809	

^†^ The Kernel-based Regularized Least Squares (KRLS) are used and the average of the pointwise marginal effects are reported. ** *p* < 0.01, * *p* < 0.05. The dependent variable is a global life satisfaction measure—the Satisfaction with Life Scale (SWLS). Severn life domains were drawn from the Personal Wellbeing Index (PWI), and they were rescaled on a 0–1 scale. Other than what is reported, covariates in all models also include a set of gender, age, education, and country dummies. All continuous variables were standardized before entering the regression analyses. M, mean; SD, standard deviation; SE, standard error.

## Data Availability

The MIC dataset is available by applying to: https://www.monash.edu/business/che/aqol/mic (accessed on 18 August 2018).

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
