# Peer review of "Measuring the Wellbeing of Cancer Patients with Generic and Disease-Specific Instruments"

_cancers, 2023, doi:10.3390/cancers15041351_

Round 1

Reviewer 1 Report

First, congratulations on the work done. I think it is an investigation that can be of great interest and impact on the readers of the journal. However, I would like the authors to review and make some minor changes:

- In the introduction, I think it could be clarifier, before we get to lines 57-58 and as has been done in the abstract, the authors should comment that measures referred to in the article are welfare measures that are often used for cancer patients, among other diseases.

- In addition, in the results it would be advisable an analysis of the roof/floor effect of the items and a reliability analysis of the items of the scales and subscales studied.

Author Response

Many thanks for your helpful feedback.

Point 1. - In the introduction, I think it could be clarifier, before we get to lines 57-58 and as has been done in the abstract, the authors should comment that measures referred to in the article are welfare measures that are often used for cancer patients, among other diseases.

Response: We have enriched the introduction following the reviewer’s suggestion.

Point 2. - In addition, in the results it would be advisable an analysis of the roof/floor effect of the items and a reliability analysis of the items of the scales and subscales studied.

Response: We have now described the % of respondents who were classified as in the best (full) and worst health/wellbeing in the Results. No instrument demonstrated ceiling/floor effect in the study sample. EQ-5D-5L has typically been reported to have a ceiling effect in the literature, whilst in this study, 11.92% of respondents would be classified as in full health according to EQ-5D-5L, and is lower than the commonly used threshold of 15%. 

Given this study mainly focused on preference-based health utility instruments and a capability instrument, in which one item represents each dimension, we did not study the reliability of these instruments. We have enriched the limitation section that test-retest reliability cannot be explored from the current study.

Reviewer 2 Report

Thank you for the opportunity it review this article. This is an important topic and contribution to the field. 

Overall, this paper is well-written, concise and provides an important contribution to the measurement and economic evaluation of the well-being of people living with cancer. The introduction provides a valid rationale for the comparison of measures.

Given the focus in the introduction on capturing the full experience of the person living with cancer when conducting an economic evaluation, it would be ideal to include an aim to recommend which measure or combination of measures would best capture all important domains of well-being. 

Based on your study, would the recommendation be to use the QLU-C10D in addition to which measure to cover missing domains? A general statement suggesting which combination would be most effective at this point in time would be useful here, as it seems that no single measure truly captures the full experience. 

Author Response

Many thanks for your suggestions.

Point 1. Given the focus in the introduction on capturing the full experience of the person living with cancer when conducting an economic evaluation, it would be ideal to include an aim to recommend which measure or combination of measures would best capture all important domains of well-being. 

Based on your study, would the recommendation be to use the QLU-C10D in addition to which measure to cover missing domains? A general statement suggesting which combination would be most effective at this point in time would be useful here, as it seems that no single measure truly captures the full experience. 

Response: We have enriched the Conclusions section to include our suggestions. We did not specifically list this as a key aim of the paper because we feel it would be better to fully explore this aim when accessing a longitudinal data set and exploring the responsiveness of different instruments, which is a limitation of this study.

Reviewer 3 Report

Thank you for your submission. This is a relevant paper with an interesting methodology. Introduction: I would suggest providing some short/easy examples of the types of questions in some of the instruments to help the reader understand what are the differences between generic HSU and SWB at a glance. Please explain eudemonia as it might not be a well-known term in all circles. It would be useful to graph the information about Subjective well-being and capability well-being instruments in a Table like you did with HSU instruments Results: You state that "The mean SWB scores (varied from 0.56 to 0.63) were consistently lower than the mean capability wellbeing score (0.81)" Yet in Table 2, ICECAP-A scored 0.81. Please explain. Clear tables Discussion: The statement: "It can be argued that SWB may not be interchangeable with HRQoL when considering outcomes in cancer" Is very relevant and should be highlighted in the conclusion. It would also benefit from a clearer "relevance to practice/research" paragraph.

Author Response

Many thanks for your comments.

Point 1. Introduction: I would suggest providing some short/easy examples of the types of questions in some of the instruments to help the reader understand what are the differences between generic HSU and SWB at a glance. 

Response: We have added some example questions when introducing the instruments in Section 2.2.

Point 2. Please explain eudemonia as it might not be a well-known term in all circles. 

Response: We have added the explanation when the first time uses this term in the Introduction.

Point 3. It would be useful to graph the information about Subjective well-being and capability well-being instruments in a Table like you did with HSU instruments 

Response: We have enriched the introduction in the text (Section 2.2.2) instead of adding a new table because there are fewer overlaps on the items (except those global life satisfaction questions) across instruments.

Point 4. Results: You state that "The mean SWB scores (varied from 0.56 to 0.63) were consistently lower than the mean capability wellbeing score (0.81)" Yet in Table 2, ICECAP-A scored 0.81. Please explain. Clear tables 

Response: Capability wellbeing is measured using ICECAP-A. We have clarified this in Table 2 and the text.

Point 5. Discussion: The statement: "It can be argued that SWB may not be interchangeable with HRQoL when considering outcomes in cancer" Is very relevant and should be highlighted in the conclusion. 

Response: We have enriched the discussion.

Point 6. It would also benefit from a clearer "relevance to practice/research" paragraph.

Response: Thank you for the suggestion. We hope the revised conclusion section would satisfy this suggestion.